# *Rubus occidentalis* and Ellagic Acid Affect the Contractility of Penile Corpus Cavernosum Smooth Muscle through the Nitric Oxide-Cyclic Guanosine Monophosphate and Cyclic Adenosine 3′,5′-Monophosphate Signaling Pathway

**DOI:** 10.3390/jcm11102947

**Published:** 2022-05-23

**Authors:** Keshab Kumar Karna, Bo-Ram Choi, Chul-Young Kim, Hye-Kyung Kim, Jong-Kwan Park

**Affiliations:** 1American Center for Reproductive Medicine, Cleveland Clinic, Cleveland, OH 44195, USA; karnakeshab@gmail.com; 2Jaseng Spine and Joint Research Institute, Jaseng Medical Foundation, Seoul 06110, Korea; bohosun@hanmail.net; 3Department of Pharmacy, Hanyang University, Ansan 15588, Korea; chulykim@hanyang.ac.kr; 4Department of Pharmacy, Kyungsung University, Busan 48434, Korea; 5Department of Urology, Institute for Medical Sciences, Jeonbuk National University Medical School, Jeonju 54907, Korea; 6Biomedical Research Institute and Clinical Trial Center for Medical Device, Jeonbuk National University Hospital, Jeonju 54907, Korea

**Keywords:** *Rubus occidentalis*, ellagic acid, erectile dysfunction, udenafil, rolipram, nitric oxide, cyclic guanosine monophosphate, cyclic adenosine 3′,5′-monophosphate

## Abstract

The present study was designed to evaluate the relaxation effect of *Rubus occidentalis* (RO) and ellagic acid (EA) on rabbit penile corpus cavernosum smooth muscle (PCCSM). Rabbit PCCSM was treated with ROE or EA after preincubation with nitric oxide synthase (NOS), guanylate cyclase (GC), adenylyl cyclase (AC) or protein kinase A (PKA) blocker. Cyclic nucleotides in the perfusate were analyzed using radioimmunoassay (RIA). Subsequently, perfused PCCSMs were subjected to analysis to evaluate the expression level of endothelial nitric oxide synthase (eNOS) and neuronal nitric oxide synthase (nNOS). The interaction of ROE or EA with phosphodiesterase (PDE) 5 and PDE4 inhibitors, such as udenafil (UDE) and rolipram (ROL), were also evaluated. Both ROE and EA relaxed the PCCSM in a concentration-dependent manner. Coincubation of ROE or EA with NOS, GC, AC, or PKA blocker significantly decreased the ROE- and EA-induced relaxation. Pretreatment of ROE and EA significantly upregulated the cyclic guanosine monophosphate (cGMP), cyclic adenosine 3′,5′-monophosphate (cAMP), and eNOS levels in the perfused PCCSM. Furthermore, the treatment of ROE and EA markedly increased the UDE- and ROL-induced relaxation of the PCCSM. In conclusion, ROE and EA induced PCCSM relaxation by activating the nitric oxide (NO)-cGMp and cAMp signaling pathways and may have a synergistic action to improve erectile function.

## 1. Introduction

Erectile dysfunction (ED) is defined as the consistent inability to achieve or maintain a penile erection that is adequate for sexual intercourse [1]. According to the Massachusetts Male Aging Study, 52% of men aged 40–70 years are affected by some form of ED [2]. In 1995, it was estimated that 152 million men were suffering from ED worldwide, with this number being expected to increase to 322 million men by 2025 [3].

Penile erection is a complex neurovascular process in which nitric oxide (NO) plays an indispensable role [4]. In response to sexual stimulation, NO is synthesized by nitric oxide synthase (NOS) and activates guanylate cyclase (GC) to enhance the synthesis of cyclic guanosine monophosphate (cGMP) in smooth muscle cells [5]. This increase in cGMp activates protein kinase G, which induces potassium channels to open; this results in a decline in intracellular calcium that ultimately leads to corpus cavernosum smooth muscle relaxation, which, in turn, triggers penile erection [6]. Additionally, an intracellular secondary messenger, namely, adenylyl cyclase (AC), catalyzes the accumulation of intracellular cyclic adenosine 3′,5′-monophosphate (cAMP), which acts on calcium channels to reduce intracellular calcium as a secondary pathway, leading to corpus cavernosum smooth muscle relaxation [5].

ED can be caused by multiple factors, including vasculogenic, neurogenic, endocrinological, medication-related, and psychogenic factors [7]. Research identified that the development of ED mainly originates due to the dysfunction of the NO that increases the contractility of smooth muscle, upregulates oxidative stress, and lowers neuronal NO-dependent relaxation of the penis corpus cavernosum [3]. NO in penile tissue is produced by three different isoforms of NOS: nNOS, eNOS, and inducible NOS (iNOS). Isoforms nNOS and eNOS are involved in the production of NO in the non-adrenergic-non-cholinergic (NANC) nerve and penile vascular system [4].

Oral phosphodiesterase (PDE) 5 inhibitors are an effective and proven first-line therapy for ED [8]. PDE5 inhibitors are predominant PDEs in the corpus cavernosum and the catalytic site of PDE5 degrades cGMp [5]. Inhibition of PDE5 potentiates the endogenous cGMp to prolong penile erection. A study in human and animal models showed that PDE5 inhibitors improve erectile function partly by increasing the amount of NO [6]. Additionally, the PDE4 inhibitor (Rolipram) is a relatively selective inhibitor for cAMp specific PDE4 isoenzyme. The PDE4 inhibitor induces an elevation of cAMp, which was reported to relax penile corpus cavernosum smooth muscle (PCCSM) in rabbit [5]. However, these drugs have adverse side effects, such as headache, dizziness, facial flushing, dyspepsia, blue vision, back pain, priapism, and cardiac arrhythmia [9]. Moreover, several studies showed that patients with diabetes-induced ED do not respond adequately to PDE5 inhibitors [10]. Therefore, potential supplements with a relaxant effect that may also have favorable effects on ED need to be identified.

*Rubus occidentalis* (RO), commonly known as black raspberry, has been cultivated for centuries as a food supplement and traditional medicine in Korea and China [11]. The fruit of RO has favorable effects on dyslipidemia, hypertension, obesity, rheumatism, indigestion, and diabetes [12]. Regarding its biological and therapeutic activities, RO is reported to not only exert antiproliferative, antioxidant, anti-inflammatory, anti-cancer, and antimicrobial effects but also enhances the function of sexual organs [13]. The major compounds in RO are anthocyanins, resveratrol, quercetin, and ellagic acid (EA) [14]. In a previous study, EA exerted a relaxant effect on the vascular endothelium to attenuate hypertension in rats [15]. However, little is known about the molecular mechanism underlying this relaxant effect of RO extract (ROE), especially that of the major compound EA; whether ROE can be used to treat ED is also unknown.

The current study was designed to assess whether ROE and EA have a relaxant effect on rabbit PCCSM and to elucidate the underlying molecular mechanism. The additive effects of ROE and EA on PDE5- or PDE4-induced PCCSM relaxation was also analyzed.

## 2. Materials and Methods

### 2.1. Chemicals and Reagents

L-phenylephrine (PE), Nω-nitro-L-arginine methyl ester hydrochloride (L-NAME), 1H-[1,2,4]oxadiazolo[4,3-a]quinoxalin-1-one (ODQ), MDL 12330A hydrochloride (MDL 12330A), H-89 dihydrochloride hydrate (H-89), dimethyl sulfoxide, ellagic acid (EA), and rolipram (ROL) were purchased from Sigma-Aldrich (St. Louis, MO, USA). Udenafil (UDE) was provided by Dong-A ST Company (Seoul, Korea). All other chemicals were of analytical grade and purchased from standard suppliers.

### 2.2. Preparation of Crude Material

Unripe and dried RO fruits were obtained from a herbal medicine market (Gochang, Korea), and their identity was confirmed by Professor Chul Young Kim of the College of Pharmacy, Hanyang University (Ansan, Korea). A voucher specimen (accession number RO-001) was deposited in the herbarium of the College of Pharmacy, Hanyang University. The unripe RO fruits (2 kg) were pulverized and extracted three times with 50% ethanol for 3 h under reflux. The concentrated ROE was dissolved in HEPES buffer before use.

### 2.3. Animals and Experimental Protocol

All animal handling and experimental procedures were approved by the Jeonbuk National University Hospital Laboratory Animal Centre, South Korea, and conducted in accordance with the Guidelines for the Care and Use of Laboratory Animals, with prior approval obtained from the Institutional Animal Care and Use Committee of the Laboratory Animal Center (cuh-IACUC-2016-12). Healthy, sexually mature New Zealand White rabbits weighing 2.5–3.0 kg were maintained under a 12 h light/12 h dark cycle and controlled temperature (18 ± 1 °C) and humidity (62 ± 5%) conditions, with water and food provided ad libitum. After 1 week of acclimatization in our laboratory environment, the rabbits were anesthetized intravenously with 50 mg/kg ketamine and 25 mg/kg xylazine hydrochloride and then exsanguinated. The penis of each rabbit was excised rapidly and strips of PCCSM (1.5 × 1.5 × 7 mm) were prepared as described previously [16]. Each strip of rabbit PCCSM was mounted vertically in an organ chamber via cannulation with a PE50 polyethylene tube. The PCCSM strips were then perfused with ROE and EA for 2 h before the cGMp and cAMp levels were analyzed. Other penis samples were perfused with ROE or EA for 2 h for Western blot analysis of the expression levels of endothelial (eNOS) and neuronal NOS (nNOS).

### 2.4. Organ Bath Study Protocol

PCCSM tension was measured as described previously [17]. Briefly, each rabbit PCCSM strip was mounted and equilibrated for 60 min, with an adjustment of the stretch level to a resting tension of 1 g. Then, PE (10^−5^ M) was added to adjust the maximum contractile tension. Once a stable contractile tension was attained, the relaxant effects of ROE and EA were evaluated at final concentrations of 1.0–4.0 mg/mL and 0.1–100 mM, respectively. The dosages of ROE and EA were selected based on a previous study [18]. To elucidate the relaxation mechanism potentially mediated by ROE and EA, we analyzed the effects of ROE and EA on PE-induced tone by comparing the relaxant effect before and after incubating the PCCSM tissue with L-NAME (1 mM), ODQ (0.01 mM), MDL 12330A (0.01 mM), or H-89 (0.1 mM) for 30 min to block NOS, GC, AC, and protein kinase A (PKA), respectively.

### 2.5. Measurement of cAMp and cGMp Concentrations via a Radioimmunoassay

Concentrations of cGMp and cAMp in perfusate were measured as described previously [18]. Briefly, 100 μL of perfusate was first mixed with 300 μL of trichloroacetic acid for 15 min at 37 °C. The sample was then centrifuged at 3000× *g* for 10 min at 4 °C. The supernatant (100 μL) was extracted with water-saturated diethyl ether three times before drying in a SpeedVac concentrator. The sample was next resuspended with sodium acetate buffer (100 μL) and assayed for cAMp and cGMp using a commercially available radioimmunoassay kit [18]. Levels of cGMp and cAMp are expressed as femtomoles per milligram (fmol/mg) of PCCSM tissue.

### 2.6. Western Blot Analysis

Proteins isolated from ROE- and EA-perfused PCCSM tissue were analyzed using Western blotting, as described previously [17]. Specifically, the levels of eNOS and nNOS in PCCSM tissue were determined. Briefly, 20 μg of lysate proteins from each sample was denatured at 95 °C for 5 min and separated using 8% sodium dodecyl sulfate–polyacrylamide gel electrophoresis, after which, the proteins were electro-transferred to a polyvinylidene fluoride membrane (#1620177; Bio-Rad, Hercules, CA, USA). The membrane was blocked for 1 h at 4 °C in Tris-buffered saline containing 0.05% Tween 20 (TBS-T; pH 7.2) and 5% non-fat milk before incubation with primary antibodies against eNOS and nNOS (catalog numbers 610298 and 610311, respectively; BD Biosciences, Franklin Lakes, NJ, USA) in TBS-T containing 5% non-fat milk overnight at 4 °C. After three washes in 0.05% TBS-T at 10 min intervals, the blot was further incubated with 1:5000-diluted goat anti-rabbit immunoglobin G antibody (G-21234; Zymed Laboratories, San Francisco, CA, USA) for 1 h at room temperature. Following three washes with TBS-T (at 10 min intervals), specific protein band signals were detected using an enhanced chemiluminescence (ECL) kit (Amersham Bioscience, Piscataway, NJ, USA), and images were acquired using an ECL visualization system (Vilber Lourmat, Collégien, France). β-actin was used as the protein loading control.

### 2.7. Assessment of Interactive Effects between ROE or EA with UDE or ROL on PCCSM Tension

PCCSM tissue was precontracted with PE (10^−5^ M), followed by exposure to 0.1 µM UDE or 1 µM ROL for 30 min before 3 mg/mL ROE or 10 mM EA was added to the organ chamber. PCCSM tissue precontracted with PE (10^−5^ M) was also preincubated with ROE or EA before the subsequent addition of UDE or ROL.

### 2.8. Statistical Analysis

The control submaximal PCCSM contractile response induced by PE was considered to correspond to 100%, and responses to ROE and EA were expressed as percentages of this value. All results are expressed as the means ± standard deviations, with n representing the number of PCCSM samples examined in each group. One-way analysis of variance followed by Tukey’s post hoc test and Student’s paired t-test, conducted using SPSS software (version 22.0; IBM Corp., Armonk, NY, USA), were used to determine whether differences were significant. *p* < 0.05 was considered to indicate statistical significance.

## 3. Results

### 3.1. Determination of the EA Content in the Crude Extract

The EA content in the crude extract was measured using high-performance liquid chromatography (HPLC; Figure 1). A 1260 HPLC system (Agilent, Santa Clara, CA, USA) with a Shiseido Capcell Pak column (C18 UG120, 150 mm × 4.6 mm, 5 μm; Shiseido Co., Ltd., Tokyo, Japan) was used. The mobile phase flow rate of 1 mL/min was controlled using binary pumps, and the temperature was set to 40 °C. The mobile phase consisted of acetonitrile and water, with an elution gradient of 10–30% acetonitrile over 0–20 min. The column was equilibrated for 15 min before each injection, and the injection volume was 10 μL. The effluent was monitored at 254 nm with an ultraviolet (UV) spectrum range of 200–500 nm. EA was eluted at 10.38 min. Compound identification was based on the HPLC retention time and UV spectrum of each compound.

### 3.2. Cumulative Effects of ROE and EA on PCCSM Tissue

The relaxant effect of ROE is presented in Figure 2. ROE induced PCCSM relaxation in a concentration-dependent manner, with the maximum relaxant effect obtained at 4 mg/mL. Pretreatment with ODQ, L-NAME, MDL 12330A, or H-89 significantly decreased the relaxation of PCCSM tissue caused by the ROE application (*p* < 0.05). Specifically, the relaxant effect was blocked at ROE concentrations of 3 mg/mL (*p* < 0.05) and 4 mg/mL (*p* < 0.01) by ODQ, L-NAME, MDL 12330A, and H-89 (Figure 2A–D).

The relaxant effect of EA on PCCSM tissue is summarized in Figure 3. EA also induced PCCSM relaxation in a concentration-dependent manner, with the maximum relaxant effect obtained at 100 mM. This effect was significantly inhibited by pretreatment with ODQ, L-NAME, MDL 12330A, or H-89 (*p* < 0.05). Specifically, the relaxant effect was blocked at EA concentrations of 10 mM (*p* < 0.05) and 100 mM (*p* < 0.01) by ODQ L-NAME, MDL 12330A, and H-89 (Figure 3A–D).

### 3.3. Effects of ROE and EA on cAMp and cGMp Concentrations in Perfusate

The effect of ROE on cAMp and cGMp levels in perfusate is presented in Table 1. The yields of both cAMp and cGMp in perfusate increased significantly upon treatment with ROE in a concentration-dependent manner (*p* < 0.05). The highest yields of cGMp and cAMp were obtained with 4 mg/mL ROE, with maximum values of 1,135.82 ± 78.94 and 173.76 ± 5.77 fmol/mg, respectively. Similarly, the yields of cAMp and cGMp increased upon treatment with EA in a concentration-dependent manner (Table 2). The most potent concentration was 100 μM, which produced maximum yields of 1209.52 ± 27.53 and 172.32 ± 2.91 fmol/mg, respectively (Table 2).

### 3.4. Expression of eNOS and nNOS in PCCSM Tissue

The levels of eNOS and nNOS observed in PCCSM tissue are summarized in Figure 4. Both ROE and EA increased the eNOS level significantly compared with the control (*p* < 0.01). Treatment with ROE led to a significant increase in nNOS expression (*p* < 0.05), whereas treatment with EA only resulted in a slight increase. Co-treatment with L-NAME and ROE or EA resulted in a significant decrease in the eNOS level compared with incubation with ROE alone (*p* < 0.001). Pretreatment with L-NAME decreased the nNOS expression induced by ROE and EA. However, the differences were not significant.

### 3.5. Effect of ROE on PCCSM Preincubated with a PDE4 or -5 Inhibitor

The relaxant effect of a single dose of ROE (3 mg/mL) on PE-treated PCCSM, also incubated with a single dose of UDE (0.1 µM) or ROL (1 µM), is presented in Figure 5. UDE or ROE alone effectively improved the relaxation of PE-treated PCCSM by 11.47 ± 1.09% and 22.21 ± 3.87%, respectively (Figure 5A). UDE and ROE co-treatment also improved the PCCSM relaxation by 34.54 ± 2.97% in tissue preincubated with UDE and 39.79 ± 1.35% in tissue preincubated with ROE. Co-treatment with ROE improved the UDE-induced relaxation significantly (*p* < 0.05), especially pretreatment with ROE (*p* < 0.01). ROL or ROE alone effectively improved the relaxation of the PE-treated PCCSM by 16.25 ± 1.76% and 22.97 ± 3.35%, respectively (Figure 5B). When ROL was combined with ROE, the PCCSM relaxation improved by 37.74 ± 3.15% after preincubation with ROL and 40.08 ± 2.27% after preincubation with ROE. Co-treatment with ROE improved the ROL-induced relaxation significantly (*p* < 0.05), especially pretreatment with ROE (*p* < 0.01). Treatment with ROE enhanced the UDE- and ROL-induced relaxation by more than twofold (*p* < 0.01) (Figure 5).

### 3.6. Effect of EA on PCCSM Preincubated with a PDE4 or -5 Inhibitor

The relaxant effect of a single dose of EA (10 mM) on PE-treated PCCSM that was also incubated with a single dose of UDE (0.1 µM) or ROL (1 µM) is presented in Figure 6. UDE or EA alone effectively improved the relaxation of PE-treated PCCSM by 12.02 ± 2.40% and 23.50 ± 2.93%, respectively (Figure 6A). UDE and EA in combination improved the PCCSM relaxation by 37.95 ± 2.57% after preincubation with UDE and 39.50 ± 1.57% after preincubation with EA. Co-treatment with EA improved UDE-induced relaxation significantly (*p* < 0.05); in particular, marked improvement was observed with pretreatment with EA (*p* < 0.01). ROL or EA alone effectively improved the relaxation of PE-treated PCCSM by 15.68 ± 2.09% and 27.80 ± 3.05%, respectively (Figure 6B). Combined treatment with ROL and EA improved the PCCSM relaxation by 40.10 ± 1.55% after preincubation with ROL and 39.52 ± 3.62% after preincubation with EA. Co-treatment with EA improved the ROL-induced relaxation significantly (*p* < 0.05), with an especially marked improvement observed after EA pretreatment (*p* < 0.01). Treatment with EA enhanced the UDE- and ROL-induced relaxation by more than twofold (*p* < 0.01) (Figure 6).

## 4. Discussion

In this study, we provided evidence that ROE and one of its major compounds, namely, EA, exerted potent relaxant effects in a concentration-dependent manner on rabbit PCCSM tissue. Dose-responsive increases in cAMp and cGMp were also observed upon exposure to ROE and EA. Furthermore, ROE and EA increased NO production and exhibited additive effects with UDE or ROL in rabbit PCCSM tissue. The results implied that the PCCSM relaxation induced by ROE or EA was mediated by the NO-cGMp and cAMp pathways (Figure 7).

Many natural products, such as those from *Cuscuta chinensis*, *Schisandra chinensis*, *Epimedium brevicornum*, *Artemisia capillaris*, and *Scutellaria baicalensis*, were reported to regulate NO metabolism in PCCSM in in vivo models [17,18,19,20,21]. To our knowledge, this is the first study to show that a RO product can exert a relaxant effect via the NO-cGMp and cAMp pathways. EA, which is a polyphenol isolated from RO, was able to induce relaxation in a dose-dependent manner at concentrations of 0.1–100 mM. In a rat model, treatment with EA induced PCCSM relaxation and suppressed diabetes-induced ED [22]. Here, a similar relaxant effect was observed after preincubation with EA. Treatment with NOS, GC, AC, and PKA blockers significantly reduced ROE- and EA-induced relaxation, indicating that ROE and EA enhanced PCCSM relaxation by activating the NO-cGMp and cAMp signaling pathways.

Pretreatment with ROE or EA led to concentration-dependent increases in the cAMp and cGMp levels in the perfusate. An extract of another *Rubus* species, namely, *R. coreanus*, was reported to have a relaxant effect on PCCSM tissue; this was mediated by the upregulation of cGMp and cAMP, which is consistent with the current findings [23]. The cGMp and cAMp signaling pathways play pivotal roles in normal erectile tissue function by reducing the tone of the PCCSM [24]. The increased cGMp and cAMp induced by ROE and EA may have a therapeutic effect on ED involving the NO-cGMp and cAMp signaling pathways.

NO, which is a potent PCCSM relaxant, is produced by NOS, and the relaxation process is mediated by cGMp [25]. Three NOS isoforms exist in the penis: nNOS, eNOS, and inducible NOS [26]. Only eNOS and nNOS are crucial for initiating and maintaining an erection [27]. Preincubation with ROE or EA increased the eNOS and nNOS levels in PCCSM tissue, whereas PCCSM relaxation induced by ROE or EA was strongly inhibited by L-NAME. L-NAME is a selective eNOS inhibitor that prevents the generation of NO by eNOS from L-arginine in endothelial cells [28]. Thus, it appears that the relaxant effects of ROE and EA involve NO and cGMP.

In recent years, oral PDE5 inhibitors, such as UDE, have been commonly used to treat patients with ED in Korea [29]. PDE5 inhibitors were shown to improve penile erection in approximately 60% of ED patients [30]. However, although PDE5 inhibitors are the first-line treatment for ED, these medications are associated with adverse side effects. Therefore, it is important to identify potential supplementary drugs that could be used in combination with PDE4 or -5 inhibitors to effectively treat ED with fewer side effects. Our results showed that ROE or EA, in combination with UDE, enhanced the relaxation of PCCSM tissue. Thus, ROE and EA may improve ED in patients who do not respond to PDE5 inhibitors. PDE4 inhibitors, such as ROL, exhibit pharmacological activities that increase cAMp and intracellular calcium in vascular tissues, in turn causing PCCSM relaxation [31]. In the present study, the combination of ROE or EA with ROL also enhanced the relaxation of PCCSM tissue, implying that ROE and EA may have additive positive effects on penile erection.

## 5. Conclusions

In conclusion, ROE and its active component, namely, EA, exerted significant concentration-dependent relaxant effects on rabbit PCCSM and improved both UDE- and ROL-induced relaxation. ROE- and EA-induced relaxation was inhibited upon exposure to NOS, GC, AC, and PKA blockers, and treatment with ROE or EA led to concentration-dependent increases in cAMp and cGMp levels, as well as increased eNOS and nNOS expression in the PCCSM, implying that ROE and EA improve penile erection via the NO-cGMp and cAMp signaling pathways. Taken together, the results indicated that ROE and EA may serve as potential therapeutics or additional supplements for patients interested in using natural products to enhance erectile performance. Combination treatment involving ROE or EA in conjunction with PDE-4 or -5 inhibitors may also benefit ED patients who do not respond well to UDE or ROL alone.

## Figures and Tables

**Figure 1 jcm-11-02947-f001:**
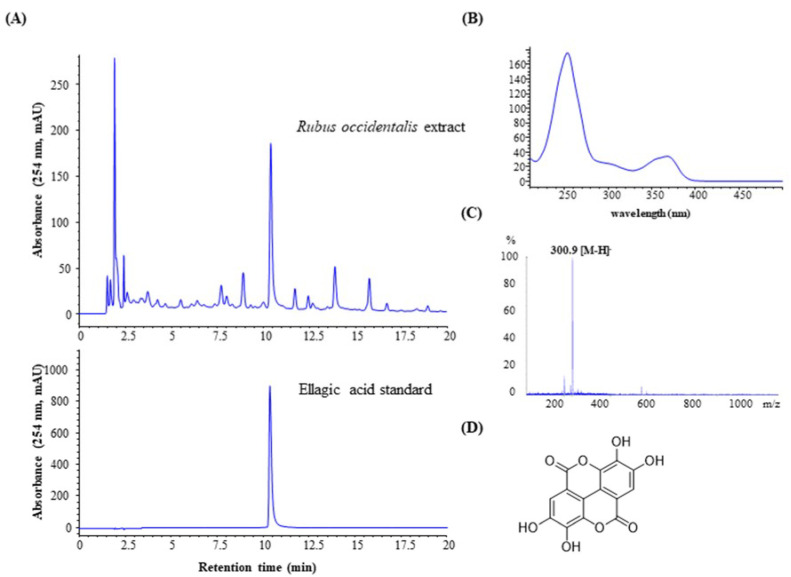
(**A**) High-performance liquid chromatography analysis of the 50% ethanol extract of unripe *Rubus occidentalis* fruit and an ellagic acid standard, (**B**) ultraviolet spectrum of ellagic acid after 10.38 min of the crude extract analysis, (**C**) mass spectrum of ellagic acid after 10.38 min of the crude extract analysis, and (**D**) the chemical structure of ellagic acid.

**Figure 2 jcm-11-02947-f002:**
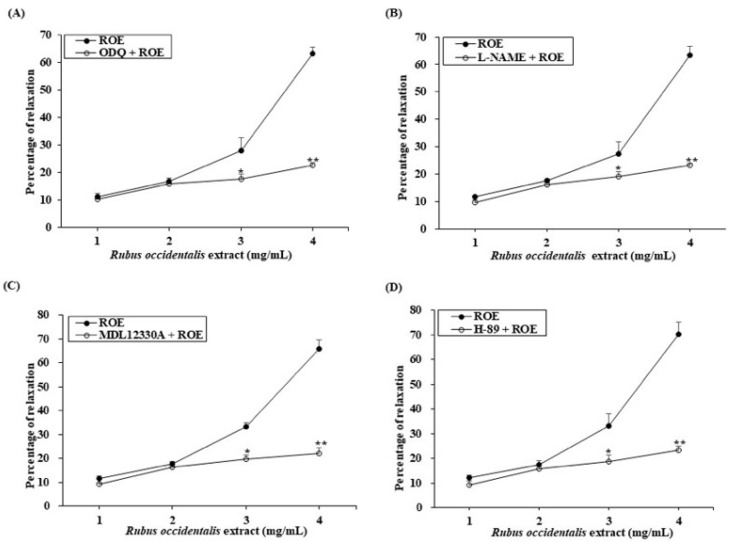
Relaxant effect of ROE on PE-induced contracted PCCSM tissue (*n* = 4). PCCSM tissue samples were contracted via exposure to PE (0.01 mM) and pretreated with ODQ (**A**) 0.01 mM), L-NAME (**B**) 1 mM), MDL 12330A (**C**) 0.01 mM), or H-89 (**D**) 0.1 mM) before treatment with ROE (1, 2, 3, or 4 mg/mL). ROE: *Rubus occidentalis* extract, PE: L-phenylephrine, PCCSM: penile corpus cavernosum smooth muscle, ODQ: 1H-[1,2,4]oxadiazolo[4,3-a]quinoxalin-1-one, L-NAME: Nω-nitro-L-arginine methyl ester hydrochloride, MDL 12330A: MDL 12330A hydrochloride, H-89: H-89 dihydrochloride hydrate. Values are presented as means ± standard deviations. * *p* < 0.05, ** *p* < 0.01 vs. treatment with ROE.

**Figure 3 jcm-11-02947-f003:**
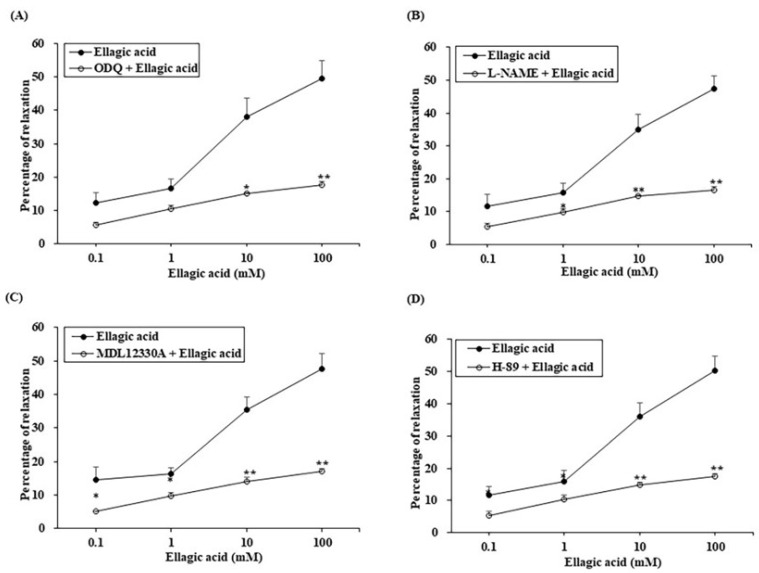
Relaxant effect of EA on PE-induced contracted PCCSM tissue (*n* = 4). PCCSM tissue samples were contracted via exposure to PE (0.01 mM) and pretreated with ODQ (**A**) 0.01 mM), L-NAME (**B**) 1 mM), MDL 12330A (**C**) 0.01 mM), or H-89 (**D**) 0.1 mM) before treatment with EA (0.1, 1, 10, or 100 mM). EA: ellagic acid, PE: L-phenylephrine, PCCSM: penile corpus cavernosum smooth muscle, ODQ: 1H-[1,2,4]oxadiazolo[4,3-a]quinoxalin-1-one, L-NAME: Nω-nitro-L-arginine methyl ester hydrochloride, MDL 12330A: MDL 12330A hydrochloride, H-89: H-89 dihydrochloride hydrate. Values are presented as means ± standard deviations. * *p* < 0.05, ** *p* < 0.01 vs. treatment with EA.

**Figure 4 jcm-11-02947-f004:**
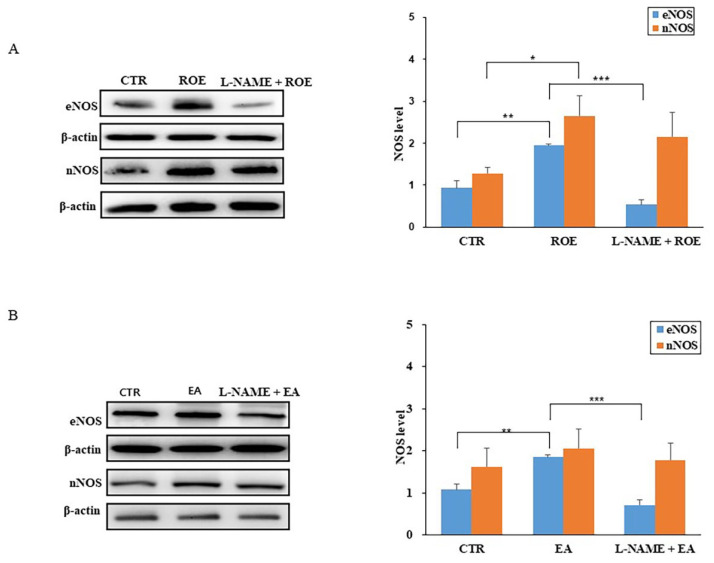
Effects of L-NAME (1 mM) on ROE- (**A**) 3 mg/mL) or EA-induced (**B**) 10 mM) eNOS and nNOS proteins in the PCCSM, as revealed using Western blot analysis (*n* = 4). L-NAME: Nω-nitro-L-arginine methyl ester hydrochloride, ROE: *Rubus occidentalis* extract, EA: ellagic acid, eNOS: endothelial nitric oxide synthase, nNOS: neuronal nitric oxide synthase, PCCSM: penile corpus cavernosum smooth muscle. Values are presented as means ± standard deviations. * *p* < 0.05, ** *p* < 0.01, *** *p* < 0.001.

**Figure 5 jcm-11-02947-f005:**
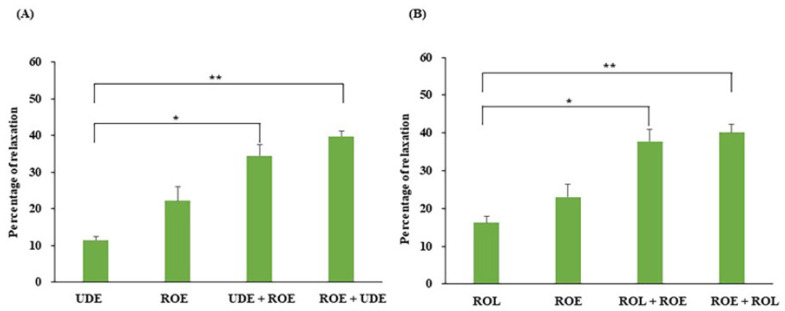
Interaction between ROE (3 mg/mL) and UDE (**A**) 0.1 µM) or ROL (**B**) 1 µM) in the context of relaxation of the PCCSM (*n* = 4). ROE: *Rubus occidentalis* extract, UDE: udenafil, ROL: rolipram, PCCSM: penile corpus cavernosum smooth muscle, UDE + ROE: ROE treatment after pretreatment with UDE, ROE + UDE: UDE treatment after pretreatment with ROE, ROL + ROE: ROE treatment after pretreatment with ROL, ROE + ROL: ROL treatment after pretreatment with ROE. Values are presented as means ± standard deviations. * *p* < 0.05, ** *p* < 0.001 vs. treatment with UDE or ROL.

**Figure 6 jcm-11-02947-f006:**
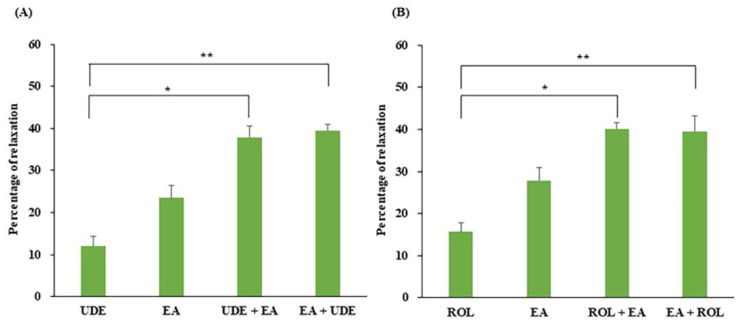
Interaction between EA (10 mM) and UDE (**A**) 0.1 µM) or ROL (**B**) 1 µM) in the context of relaxation of the PCCSM (*n* = 4). EA: ellagic acid, UDE: udenafil, ROL: rolipram, PCCSM: penile corpus cavernosum smooth muscle, UDE + EA: EA treatment after pretreatment with UDE, EA + UDE: UDE after pretreatment with EA, ROL + EA: EA treatment after pretreatment with ROL, EA + ROL: ROL treatment after pretreatment with EA. Values are presented as means ± standard deviations. * *p* < 0.01, ** *p* < 0.001 vs. treatment with UDE or ROL.

**Figure 7 jcm-11-02947-f007:**
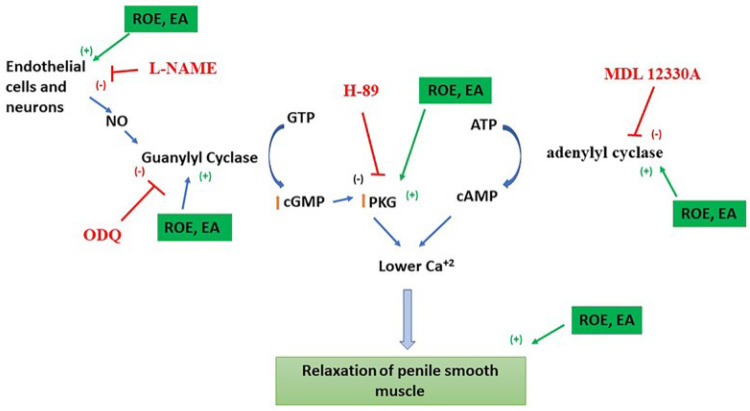
Schematic diagram showing the relaxant effects of ROE and EA on rabbit PCCSM, as mediated by the NO-cGMp and cAMp signaling pathways. ROE: *Rubus occidentalis* extract, EA: ellagic acid, L-NAME: Nω-nitro-L-arginine methyl ester hydrochloride, ODQ: 1H-[1,2,4]oxadiazolo[4,3-a]quinoxalin-1-one, MDL 12330A: MDL 12330A hydrochloride, H-89: H-89 dihydrochloride hydrate, NO: nitric oxide, GTP: guanosine triphosphate, cGMP: cyclic guanosine monophosphate, PKG: protein kinase G, ATP: adenosine triphosphate, cAMP: adenosine 3′,5′-cyclic monophosphate.

**Table 1 jcm-11-02947-t001:** Cyclic nucleotides in perfusate with ROE (*n* = 4).

Samples	Concentrations (mg/mL)	Cyclic Nucleotides	
		cAMp (fmol/mg)	cGMp (fmol/mg)
ROE (mg/mL)	Control	957.25 ± 46.34	121.86 ± 4.51
	1	1032.09 ± 13.44	154.36 ± 7.87 *
	2	1036.84 ± 26.25	157.35 ± 7.06 **
	3	1069.60 ± 40.58 *	162.86 ± 4.91 **
	4	1135.82 ± 78.94 *	173.76 ± 5.77 **

Values are presented as mean ± standard deviations. ROE: *Rubus occidentalis* extract, cAMP: cyclic adenosine monophosphate, cGMP: cyclic guanosine monophosphate. *, **: Statistically significant from control concentrations, where * *p <* 0.01 and ** *p <* 0.001.

**Table 2 jcm-11-02947-t002:** Cyclic nucleotides in perfusate with EA (*n* = 4).

Samples	Concentrations (µM)	Cyclic Nucleotides	
		cAMp (fmol/mg)	cGMp (fmol/mg)
EA (µM)	Control	947.09 ± 38.33	114.45 ± 0.86
	0.1	1034.10 ± 14.93 *	152.27 ± 6.95 *
	1	1037.60 ± 26.41 *	156.68 ± 10.35 *
	10	1065.52 ± 42.43 *	163.26 ± 5.70 **
	100	1209.52 ± 27.53 **	172.32 ± 2.91 **

Values are presented as mean ± standard deviations. EA: ellagic acid, cAMP: cyclic adenosine monophosphate, cGMP: cyclic guanosine monophosphate. *, ** Statistically significant from control concentrations, where * *p* < 0.01 and ** *p* < 0.001.

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
