# Peer review of "Rubus occidentalis and Ellagic Acid Affect the Contractility of Penile Corpus Cavernosum Smooth Muscle through the Nitric Oxide-Cyclic Guanosine Monophosphate and Cyclic Adenosine 3′,5′-Monophosphate Signaling Pathway"

_jcm, 2022, doi:10.3390/jcm11102947_

Round 1

Reviewer 1 Report

Thank you for the opportunity to review the manuscript entitled ‘Ellagic Acid from Rubus occidentalis Affects the Contractility of Penile Corpus Cavernosum Smooth Muscle through the Nitric Oxide-Cyclic Guanosine Monophosphate Pathway’. The study concluded that ROE and EA-induced PCCSM relaxation by activating nitric oxide (NO)-cGMP and cAMP signalling pathways and may have synergistic action to improve erectile function.

This study is well designed, written and presented. However, there are some recommendations provided below to address:  

Introduction

  • Ln51: Consider a new paragraph on line 51 to separate physiology from causes/mechanisms/treatments and adverse effects; after ‘leading to corpus cavernosum smooth muscle relaxation’
  • Ln51 – Ln 58: Consider to include a comment on the role of reduced nitric oxide as a common molecular pathway (reduced NO, and NOS too). This relates directly to the study outcomes and results. Briefly include roles of PDE4 and PDE5 too here, or with the physiology, as important aspects of the proposed study aim
  • Ln53: Also mention that the PDE5 inhibitors improve erectile function partly through increasing NO too here.
  • Also consider to include and differentiate eNOS and nNOS relevant to using these as study outcomes (this is mentioned in the discussion, but may be useful in the introduction)

Methods

  • Figure 1 should be linked to results, as it is the outcome of the HPLC; please move this to the results and embed in text there.
  • Please clarify how the EA was isolated? EA was identified in the crude extract by HPLC, and the crude extract is defined as ROE and used experimentally; However, it is not clear how the EA was obtained
  • Include a justification for the dosages used for ROE and EA based on previous studies, or a comment on the basis for these concentrations to be selected
  • Statistics: Please confirm all data was parametric, as parametric tests were used.

Results

  • Please specify the concentration unit with the numbers in Table 1, instead if concentrations 1 – 4. It does appear at glance as a numbering and not a concentration without the units there.

Author Response

Reviewer reports:

Reviewer #1: (Comments to the Author):

Introduction

 Comment #1

Ln51: Consider a new paragraph on line 51 to separate physiology from causes/mechanisms/treatments and adverse effects; after ‘leading to corpus cavernosum smooth muscle relaxation’

Reply 1: Thank you for your detailed review and kind comments. As per your suggestion we have added the new paragraph after ‘leading to corpus cavernosum smooth muscle relaxation’.

Comment #2

Ln51 – Ln 58: Consider to include a comment on the role of reduced nitric oxide as a common molecular pathway (reduced NO, and NOS too). This relates directly to the study outcomes and results. Briefly include roles of PDE4 and PDE5 too here, or with the physiology, as important aspects of the proposed study aim

Reply 2: Thank you for your detailed review and kind comments. We have included the comment on the role of reduced NO. We have also included the role of PDE4 and PDE5.

Page 2, Line 55-58

Research has identified the development of ED is mainly originated due to dysfunction of NO that increases the contractility of smooth muscle, upregulate oxidative stress and lower neuronal NO dependent relaxation of penis corpus cavernosum [3].

Page 2, Line 63-65

PDE5 are predominant PDE in corpus cavernosum and catalytic site of PDE5 degrades cGMP [5]. Inhibition of PDE5 potentiate the endogenous cGMP to prolong penile erection.

Page 2, Line 66-69

Additionally, PDE4 inhibitor (Rolipram) is relatively selective inhibitor for cAMP specific PDE4 isoenzyme. PED4 inhibitor induces elevation of cAMP has been reported to relax penile corpus cavernosum smooth muscle (PCCSM) in rabbit [5].

Comment #3

Ln53: Also mention that the PDE5 inhibitors improve erectile function partly through increasing NO too here.

Reply 3: Thank you for your detailed review and kind comments. As per your comment we mention the PDE5 inhibitor improve ED partially through increasing NO.

Page 2, Line 65-66

Study in human and animal models shown that PDE5 inhibitors improve erectile function partly through increasing NO [6].

Comment #4

Also consider to include and differentiate eNOS and nNOS relevant to using these as study outcomes (this is mentioned in the discussion, but may be useful in the introduction)

 Reply 4: Thank you for your detailed review and kind comments. As per your comment we include the differentiate eNOS and nNOS relevant to using these as study outcome.

Page 2, Line 59-61

NO is penile tissue is produced by three different isoforms of NOS: nNOS, eNOS and inducible NOS (iNOS). Isoforms nNOS and eNOS are involved in production of NO in non-adrenergic-non-cholinergic (NANC) nerve and penile vascular system [4].

Methods

 Comment #5

Figure 1 should be linked to results, as it is the outcome of the HPLC; please move this to the results and embed in text there.

Reply 5: Thank you for your detailed review and kind comments. As per your suggestion we have moved the outcome of HPLC to the result.

Comment #6

Please clarify how the EA was isolated? EA was identified in the crude extract by HPLC, and the crude extract is defined as ROE and used experimentally; however, it is not clear how the EA was obtained

Reply 6: Thank you for your detailed review and kind comments. Compound identification in crude extract was based on the HPLC retention time and UV spectrum of each compound. Ellagic acid (EA) was purchased from Sigma-Aldrich. As per your suggestion we have added ellagic acid in chemicals and reagents of method on the one 94.

Comment #7

Include a justification for the dosages used for ROE and EA based on previous studies, or a comment on the basis for these concentrations to be selected

Reply 7: Thank you for your detailed review and kind comments. As per your suggestion we have added the justification on the line 137.

Comment #8

Statistics: Please confirm all data was parametric, as parametric tests were used.

Reply 8: Thank you for your detailed review and kind comments. The parametric tests were used for all data. One-way analysis of variance followed by Tukey’s post-hoc test and Student’s paired t-test was used to determine whether differences were significant.

Results

Comment #9

Please specify the concentration unit with the numbers in Table 1, instead if concentrations 1 – 4. It does appear at glance as a numbering and not a concentration without the units there.

Reply 9: Thank you for your detailed review and kind comments. As per your suggestion we have specified the concentration unit with the numbers in Table 1. We have also specified the concentration unit with the numbers in Table 2.

Reviewer 2 Report

Karna et al. presented a study to evaluate the relaxing effects of rubus occidentalis extract and ellagic acid on rabbit corpus cavernosum smooth muscle. The article is well written, the results are clearly illustrated, and the discussion is well structured. The topic, moreover, is of definite interest. Therefore, I have only a few minor comments:
- The title does not clearly illustrate the results of the study, since not only NO-cGMP but also the cAMP pathway appears to be involved. Also, it would be better to write "Rubus occidentalis and ellagic acid," rather than "ellagic acid from rubus occidentalis"
- In the abstract, ROE should refer to Rubus occidentalis extract (add "extract" or replace the abbreviation with RO). The abbreviation for PDE5 should also be added.
- Page 2, line 7: I suggest changing "endocrinogenic" to "endocrinological"
- All figures should be enlarged to reach the right margin of the page. In their current state they are hardly legible

Author Response

Reviewer reports:

Reviewer #2: (Comments to the Author):

Comment #1

The title does not clearly illustrate the results of the study, since not only NO-cGMP but also the cAMP pathway appears to be involved. Also, it would be better to write "Rubus occidentalis and ellagic acid," rather than "ellagic acid from rubus occidentalis"

Reply 1: Thank you for your detailed review and kind comments. As per your suggestion we have change the title of our manuscript: “Rubus occidentalis and Ellagic acid affects the contractility of penile corpus cavernosum smooth muscle through the nitric oxide-cyclic guanosine monophosphate and cyclic adenosine 3′, 5′-monophosphate signaling pathway.”

Comment #2

In the abstract, ROE should refer to Rubus occidentalis extract (add "extract" or replace the abbreviation with RO). The abbreviation for PDE5 should also be added.

Reply 2:

Thank you for your detailed review and kind comments. We have changed abbreviation with RO. As per your suggestion we have added the abbreviation for PDE5. We have also added the abbreviation for PDE4.

Comment #3

Page 2, line 7: I suggest changing "endocrinogenic" to "endocrinological"

Reply 3: Thank you for your detailed review and kind comments. As per your suggestion we have changed the word endocrinological.

Comment #4

All figures should be enlarged to reach the right margin of the page. In their current state they are hardly legible

Reply 4: Thank you for your detailed review and kind comments. As per your suggestion we have enlarged all the figures.
